# Vitamin D-Mediated Regulation of Intestinal Calcium Absorption

**DOI:** 10.3390/nu14163351

**Published:** 2022-08-16

**Authors:** James C. Fleet

**Affiliations:** Department of Nutritional Sciences, University of Texas, Austin, TX 78723, USA; james.fleet@austin.utexas.edu

**Keywords:** diet, transcellular, absorption, diffusion, intestine, homeostasis, parathyroid hormone

## Abstract

Vitamin D is a critical regulator of calcium and bone homeostasis. While vitamin D has multiple effects on bone and calcium metabolism, the regulation of intestinal calcium (Ca) absorption efficiency is a critical function for vitamin D. This is necessary for optimal bone mineralization during growth, the protection of bone in adults, and the prevention of osteoporosis. Intestinal Ca absorption is regulated by 1,25 dihydroxyvitamin D (1,25(OH)_2_ D), a hormone that activates gene transcription following binding to the intestinal vitamin D receptor (VDR). When dietary Ca intake is low, Ca absorption follows a vitamin-D-regulated, saturable pathway, but when dietary Ca intake is high, Ca absorption is predominately through a paracellular diffusion pathway. Deletion of genes that mediate vitamin D action (i.e., VDR) or production (CYP27B1) eliminates basal Ca absorption and prevents the adaptation of mice to low-Ca diets. Various physiologic or disease states modify vitamin-D-regulated intestinal absorption of Ca (enhanced during late pregnancy, reduced due to menopause and aging).

## 1. Introduction

It has now been 100 years since E.V. McCollum first identified a fat-soluble compound in food that supported bone growth and prevented rickets; he called this compound vitamin D. In the intervening century, many scientists have contributed to our understanding for how vitamin D regulates the physiology of calcium (Ca) metabolism. For example, in 1937 Nicolaysen showed that vitamin D is critical for intestinal Ca absorption [1] while later studies by Pansu et al. [2] and Sheikh et al. [3] showed that vitamin D deficiency significantly reduces intestinal Ca absorption. The critical breakthrough defining the mechanism used by vitamin D to regulate Ca metabolism came in the early 1970′s when research by Holick et al. [4] and Norman et al. [5] isolated the active metabolite of vitamin D, 1,25 dihydroxyvitamin D_3_ (1,25(OH)_2_D), from the intestine. Shortly thereafter, Brumbaugh and Haussler [6] discovered the nuclear receptor for 1,25(OH)_2_D, the vitamin D receptor (VDR), in intestinal mucosa as well. Since then, we’ve learned the detailed mechanism used by 1,25(OH)_2_D to regulate gene expression [7] while global and conditional VDR knockout mice have allow us to study the function of vitamin D in Ca/bone metabolism and in other physiologic systems. As part of this effort, my research group has shown that the single most important role for vitamin D during growth is the regulation of intestinal Ca absorption [8] but that 1,25(OH)_2_D signaling through the VDR has a broad array of target genes in the intestine [9]. Because of the central role that vitamin D plays in the regulation of intestinal Ca absorption, this review provides a critical starting point for anyone who wants to understand the physiologic importance of vitamin D.

## 2. Vitamin D Has a Critical Physiologic Role for Protecting Bone through the Regulation of Intestinal Ca Absorption

Bone mass is lost when dietary Ca intake is inadequate and so one usually thinks of bone when the term “Ca homeostasis” is used. However, Ca homeostasis is not regulated to maintain bone integrity. Instead, bone, the parathyroid gland, intestine, and kidney make up a multi-tissue axis that work together to maintain serum Ca within a narrow range (8.9–10.2 mg/dL). As a result, after a meal intestinal Ca absorption is a signal that disturbs and elevates serum Ca while bone formation and resorption, along with renal Ca excretion, respond to fluxes in serum Ca in an attempt to limit perturbations in serum Ca. This coordination was shown clearly by Bronner and Aubert who used Ca kinetics in rat models to show how the body adapts to habitual low Ca intake by increasing Ca absorption efficiency, reducing renal Ca loss, and mobilizing Ca from the bone (i.e., resorption) [10]. Pansu et al. [11] showed the impact of dietary Ca intake on intestinal Ca absorption directly when they found that feeding rats a 0.17%, low Ca diet for 5 weeks increased duodenal Ca absorption efficiency by increasing the saturable component of transport (Vmax increased 55%). Similarly, Norman et al. [12] found that in adult humans, feeding a diet with 300 mg of Ca/day for 8 weeks increased Ca absorption efficiency by 43% compared to subjects consuming 1600 mg Ca/day. These studies, and others like them, led to the identification of 1,25 dihydroxyvitamin D (1,25(OH)_2_ D) and parathyroid hormone (PTH) as the major hormonal regulators of Ca homeostasis.

When dietary Ca is habitually low, there is a transient reduction in serum Ca that is sensed at the parathyroid gland through the Ca sensing receptor (CaSR). This cell surface receptor mediates signals into the parathyroid gland to increase the production and release of PTH into the circulation—a condition called nutritional secondary hyperparathyroidism. PTH has several important functions in Ca metabolism. First, it regulates renal production of 1,25(OH)_2_ D by inducing the CYP27B1 gene that encodes the enzyme 25 hydroxyvitamin D-1α hydroxylase [13] and it suppresses expression of the CYP24A1 gene that encodes the 25 hydroxyvitamin D-24 hydroxylase [14,15]. 1,25(OH)_2_ D released from the kidney is the most important regulator of increased intestinal Ca absorption. Of course, this physiologic adaptation has limits so that if the degree of dietary Ca deprivation is too great, the adaptation of intestinal Ca absorption will not be sufficient to compensate. This case, 1,25(OH)_2_ D and PTH will both promote bone resorption by stimulating osteoclastic activity while also enhancing renal Ca reabsorption in the proximal renal tubule. Collectively, this physiological adaptation can protect serum Ca but at the expense of bone mass.

The central role for vitamin D as a regulator of intestinal Ca absorption has been known for more than 80 years [1,16]. In vitamin D deficient animals intestinal Ca absorption efficiency falls by >75% [2]. Similarly, dialysis patients with low circulating 1,25(OH)_2_ D levels also have low intestinal Ca absorption [3]. Finally, in elderly adults, secondary hyperparathyroidism can maintain serum 1,25(OH)_2_ D (and Ca absorption) until serum 25-hydroxyvitamin D (25(OH)D) levels fall to ≤10 nmol/L, at which point there is not enough substrate to convert to 1,25(OH)_2_ D [17].

A challenging concept for many people examining Ca absorption is that it is not a single process but the sum of events that occur through two routes, a transcellular, saturable pathway and non-saturable, paracellular diffusion pathway [11,18,19,20] (see Figure 1). The relative importance of these two routes depends upon a person’s habitual Ca intake. When Ca intake is low, like most adult American women [21], the saturable pathways predominates while when Ca intake is high the bulk of absorption occurs through the diffusional route. Absorption through these two routes can be modeled mathematically using a Michaelis–Menten-like equation that contains a linear component that models diffusion (Figure 1). The saturable transport pathway comprises three parts, apical membrane Ca entry, transcellular diffusion, and basolateral membrane extrusion. The apical membrane transport occurs down a concentration gradient while basolateral membrane extrusion is against a concentration gradient and requires energy [22]. The saturable pathway is present in the proximal small intestine (duodenum and jejunum), cecum, and colon [23,24,25,26,27] but is absent in the ileum [2]. Studies in rat duodenum [2] and in differentiated monolayers of the human intestinal cell line Caco-2 [28], show that 1,25(OH)_2_ D acts on the saturable transport component where it increases the Vmax (maximal absorptive capacity) but not Km (the affinity of the process for Ca). This suggests that 1,25(OH)_2_ D increases the production of intestinal Ca transporters (which we’ll discuss below) but that this increase has limits. In contrast, the passive Ca movement across the intestinal barrier occurs at ~13% of the luminal Ca level per hour in humans [20]) and is seen in all segments of the intestine. There is some evidence that the non-saturable portion of Ca absorption in the human ileum is also vitamin D sensitive; the slope of the non-saturable transport pathway is reduced in chronic renal disease patients, and it returns to normal after 1,25(OH)_2_ D injection [20].

In normal healthy adults, the Km for the saturable component of Ca absorption from the small intestine of adults is 3.3 mM, a concentration met by 265 mg Ca in a meal (calculated from data in [19,20]). As a result, when a person eats a meal with ~400 mg (1/3 the RDA for Ca), saturable Ca transport is about 60% of total Ca absorption. However, the apparent efficiency of total Ca absorption falls as the meal Ca intake level is increased and the diffusional component of absorption takes a larger role. Normally, the amount of Ca absorbed in each intestinal segment is determined by: (a) the presence of the saturable and non-saturable pathways, (b) the residence time in the segment, and (c) the solubility of Ca within the segment (e.g., lower Ca solubility at higher pH [29,30,31]) (Figure 2). Although 1,25(OH)_2_ D clearly increases intestinal Ca absorption efficiency [2,32,33], some have argued that elevated serum 25(OH)D levels might also regulate intestinal Ca absorption. This is not supported by several large, well-designed studies that show no benefit of increasing serum 25(OH)D levels beyond 50 nmol/L [34,35,36,37].

There are many studies that show adequate dietary Ca and vitamin D, and therefore total intestinal Ca absorption, is necessary for adequate bone growth. Deficiency of either Ca or vitamin D in growing children or animals causes nutritional rickets characterized by under-mineralized bone and low bone mass [39,40]. This is consistent with the concept that bone matrix cannot mineralize in the absence of mineral. However, net Ca absorption (which reflects both transport routes) is positively correlated to Ca balance in children, reflecting the critical role for Ca absorption in optimizing peak bone mass [41]. Consistent with this idea, we have shown that efficiency of Ca absorption through the saturable, vitamin-D-regulated pathway is significantly positively correlated with femoral trabecular bone volume/total volume in a genetically diverse population of 11 inbred mouse lines [42]. In addition, Patel et al. [43] reported that femur neck BMD was significantly positively correlated with Ca absorption efficiency in adult men. Several studies also indicate the high intestinal Ca absorption efficiency can protect against femoral bone loss in mice fed low Ca diets [44] or reduce the risk of osteoporotic hip fracture in women with low dietary Ca intake [45]. Collectively, these data show that both adequate Ca intake and genetically programed high intestinal Ca absorption efficiency are necessary to build and protect strong bones.

## 3. Vitamin D Effects on Intestinal Ca Absorption Are Mediated through the VDR

1,25(OH)_2_ D regulates Ca metabolism and intestinal Ca absorption by regulating gene transcription, a process that requires binding of the hormone to the Vitamin D Receptor (VDR), a nuclear receptor that is a ligand-activated transcription factor [7,46]. A number of studies have shown the critical importance of VDR for the regulation of Ca and bone metabolism. For example, children with inactivating mutations in the VDR gene (i.e., type II genetic rickets) have defects in Ca metabolism that include lower intestinal Ca absorption efficiency [47]. Similarly, VDR knockout mice have severe defects in bone growth and mineralization as well as a >70% reduction in Ca absorption efficiency [48,49]. While the gross phenotype of VDR knockout mice is abnormal bone, several lines of evidence indicate that the most important role for VDR in Ca/bone metabolism during growth is the control of intestinal Ca absorption. First, the phenotype of the intestine-specific VDR knockout mouse is identical to dietary Ca deficiency (osteomalacia, reduced serum Ca, elevated serum 1,25(OH)_2_ D levels) [50]. Conversely, feeding high Ca/high phosphate/high lactose “rescue” diets that promote passive/diffusional intestinal Ca absorption can prevent the abnormal bone and Ca metabolism phenotype of VDR knockout mice [51]. Finally, experiments from my lab showed that the VDR knockout mouse phenotype (e.g., hypocalcemia, elevated serum PTH, low bone mineral density) can be completely prevented by intestine epithelial cell-specific, transgenic expression of VDR that normalizes intestinal Ca absorption [8].

Several studies show that lower intestinal VDR levels disrupt the physiologic response to 1,25(OH)_2_D. In VDR KO mice, low level, intestine-specific transgenic VDR expression (10% of wild-type values) was insufficient to maintain normal intestinal Ca absorption [52]. Meanwhile, my lab has shown that a 50% reduction in intestinal VDR levels blunts 1,25(OH)_2_ D-regulated intestinal Ca absorption efficiency [53]. Consistent with the idea that reduced VDR function impairs intestinal Ca absorption, several studies have shown that Ca absorption efficiency is reduced in people with the longer, less transcriptionally active “f” allele of the Fok I restriction fragment length polymorphism [54,55,56]. Collectively these data support the hypothesis that variations in VDR level or function can influence vitamin-D-regulated intestinal Ca absorption as well as optimal intestinal responses to the increased serum 1,25(OH)_2_ D levels that accompany dietary Ca restriction.

## 4. Molecular Models of Ca Absorption

Ion microscopy reveals that Ca can enter at the apical membrane and flow through the absorptive epithelial cell in 20 min [57]. However, during vitamin D deficiency Ca becomes trapped in the region just below the microvilli. Treating vitamin D deficient chicks with 1,25(OH)_2_ D reverses this effect starting 2–4 h after treatment [58], consistent with the induction of gene expression mediated through the VDR. In 1986 Bronner et al. [59] critically reviewed transport data from a wide variety of well-controlled mechanistic studies, and from this analysis built the facilitated diffusion model (Figure 3, with protein distributions across the intestinal tract in Figure 2). In the first step of this model, brush border membrane uptake of Ca is mediated by an apical membrane Ca channel, which was later identified as the transient receptor potential cation channel vanilloid family member 6 (TRPV6, originally called CaT1 or ECAC2) [60]. TRPV6 gene expression is strongly regulated by 1,25(OH)_2_ D in the duodenum of mice [33,61] and in Caco-2 cells [62] and this induction is mediated by VDR binding enhancers upstream from the transcription start site [63,64]. Induction of TRPV6 mRNA precedes the increase in duodenal Ca absorption that occurs following 1,25(OH)_2_ D injection [33]. While initial studies suggested that 1,25(OH)_2_ D-mediated Ca absorption was not reduced in TRPV6 knockout mice [65,66], later studies showed that TRPV6 knockout mice [66], and mice with a D541A variant TRPV6 that inactivates Ca movement through the channel [67], had a blunted ability to increase Ca absorption in response to feeding a low Ca diet. In addition, my lab has shown that intestine-specific transgenic expression of TRPV6 increases Ca absorption efficiency and that this prevents the abnormalities in bone/Ca metabolism of VDR knockout mice [68].

An alternative model for apical membrane Ca uptake during Ca absorption is that Ca flows through the L-type Ca channel Ca_v_1.3 (Figure 3), a transporter activated by glucose-induced membrane depolarization following a meal (reviewed in [69]). However, several studies do not support a physiologic role of Ca_v_1.3 for vitamin-D-regulated Ca absorption in growing mice [70,71]. In contrast, other studies suggest that Ca_v_1.3 may have a prolactin-regulated role in transcellular Ca transport during lactation [72] and contributes to Ca absorption prior to the development of vitamin-D-regulated Ca absorption in the neonatal mouse [73].

The central player in the facilitated diffusion model is calbindin-D, a cytoplasmic Ca binding protein [59] found in intestine (the 9 kd form, calbindin D_9k_) and the kidney (the 28 kd form, calbindin D_28k_) [74] (Figure 2 and Figure 3). This was based on studies that show: (a) intestinal calbindin D_9k_ protein levels are positively correlated to Ca absorption [59], (b) intestinal calbindin D_9k_ levels are significantly reduced in vitamin D deficient animals and in VDR knockout mice [48,75], (c) 1,25(OH)_2_ D injections increase intestinal calbindin D_9k_ levels [76], and (d) theophylline-mediated inhibition of Ca binding to calbindin D_9k_ disrupts intestinal Ca absorption [77]. These observations led to the hypothesis that calbindin D acts as a ferry for intracellular Ca movement during Ca absorption [57,78]. In contrast, other studies indicate that calbindin D_9k_ is not essential for intestinal Ca absorption but may instead act as an intracellular Ca buffer that protects cells from increases in intracellular Ca during Ca absorption. For example, neither basal nor 1,25(OH)_2_ D-induced Ca absorption are reduced in calbindin-D_9k_ null mice [66,79]. In contrast, 1,25(OH)_2_ D-induced Ca absorption is reduced by 60% in calbindin-D_9k_/TRPV6 double knockout mice [66], suggesting the interaction of TRPV6 and calbindin-D_9k_ has a special role in Ca absorption. Another observation that suggests elevated calbindin levels alone are not sufficient to drive intestinal Ca absorption is that calbindin-D protein remains elevated in the intestine even after 1,25(OH)_2_ D-induced Ca absorption returns to normal in chicks [80] and mice [33]. Finally, we have observed that intestinal calbindin-D_9k_ levels increase in intestine-specific TRPV6 transgenic mice with elevated intestinal Ca absorption efficiency even in VDR knockout mice [68]. This suggests that calbindin-D_9k_ is an intracellular Ca buffer that increases in response to elevated transcellular Ca absorption and that it is not induced to act as a facilitator of transcellular Ca movement.

The final step in the facilitated diffusion model is the extrusion of Ca from the cell. This is an energy dependent process [22] mediated by the plasma membrane Ca ATPase 1b (PMCA1b) [81,82] (Figure 2 and Figure 3). Deletion of the PMCA1b gene (Atp2b1), or the 4.1R protein that stabilizes PMCA1b in the basolateral membrane, reduces basal and 1,25(OH)_2_ D-induced intestinal Ca absorption [83,84]. While some suggest that the basolateral extrusion of Ca is also be mediated by a sodium-Ca exchanger [85], disrupting the sodium gradient necessary for sodium-Ca exchange did not block duodenal Ca transport [22].

Most intestinal Ca absorption research has focused on vitamin-D-regulated saturable Ca transport but several studies have shown that vitamin D signaling can increase Ca diffusion across the jejunum and ileum [20,86]. Tudpor et al. [87] found that 1,25(OH)_2_ D induced Ca ion movement across the intestinal barrier by a solvent drag mechanism that may involve charge selectivity of the tight junction. This is similar to the role of paracellin 1 (aka claudin 16), a tight junction protein that regulates ion-specific movement of magnesium and Ca in the kidney [88]. Consistent with this idea, 1,25(OH)_2_ D treatment significantly increased claudin-2 and 12 mRNA levels in Caco-2 cells and siRNA against these claudins reduced Ca permeability across Caco-2 cell monolayers [89]. In vivo, claudin 2/claudin 12 double knockout mice (but not single KO mice) have reduced Ca absorption across the colon but not small intestine [90]. Claudin-2 and -12 expression is highest in the distal small intestine [91] but essentially absent from the duodenum. A role for claudins in paracellular Ca transport may explain why the non-saturable component of ileal Ca absorption was reduced in chronic kidney disease patients with low serum 1,25(OH)_2_ D levels [20].

Two other models have been presented to explain vitamin D-mediated, transcellular intestinal Ca absorption: vesicular transport and transcaltachia. The vesicular transport model is an alternative to the role proposed for calbindin-D as a Ca ferry during transcellular intestinal Ca absorption. This is based on the observation that 1,25(OH)_2_ D treatment increased the activity and cycling of lysosomes [92,93], that Ca accumulates within brush border membrane endosomes [94] and in lysosomes [95] during Ca absorption, and that disrupting lysosomal pH prevents lysosomal Ca accumulation and blocks Ca absorption [95,96]. While these data suggest that vesicular movement may be a legitimate pathway for uptake and movement of Ca through the intestinal epithelial cell, it is not clear what makes the vesicular transport pathway specific for Ca. Transcaltachia is a mode of Ca transport that occurs within minutes of exposing the basolateral side of enterocytes to physiologic levels of 1,25(OH)_2_ D. Transcaltachia has been directly demonstrated in the perfused chick duodenum [97]. Some data suggests transcaltachia results from 1,25(OH)_2_ D binding to a unique, alternative ligand binding pocket [98,99] in VDR within caveolae [100], i.e., a novel non-nuclear role for the receptor. Other data suggest the basolateral membrane protein mediating transcaltachia is a multi-functional Membrane Associated Rapid Response Steroid receptor (MARRS). However, while intestine-specific deletion of MARRS in mice reduced cellular 1,25(OH)_2_ D binding, disrupted 1,25(OH)_2_ D regulated Ca and phosphate uptake into enterocytes [101,102] and reduced basal Ca absorption in by 30% [103], these reports have not reported the physiological impact of MARRS deletion on bone density. Additionally, the rapid fluxes in serum 1,25(OH)_2_ D needed for transcaltachia have not been reported during the consumption of Ca-rich meals when transcaltachia would have to occur for the physiologic benefit of the process to be realized. As such, transcaltachia is not a generally accepted mechanism for vitamin-D-regulated intestinal Ca absorption.

## 5. Physiologic Regulation of Vitamin D-Mediated Intestinal Ca Absorption

As it has been described above, the major physiologic condition where vitamin D signaling is engaged to regulate intestinal Ca absorption is the habitual consumption of a low Ca diet. However, there are a number of other physiologic states that affect vitamin D metabolism or action to influence intestinal Ca absorption, i.e., growth and development, pregnancy/lactation, and aging.

The bulk of mechanistic studies on intestinal Ca absorption have been conducted in growing 2–3-month-old rodents, but recent studies in mice indicate that vitamin D-mediated Ca absorption is not important prior to weaning [73]. This is not completely surprising as VDR is not expressed prior to 14d postnatally in rodents [104,105]. Ca absorption studies in premature infants also suggests that Ca absorption during late development is vitamin D independent [106]. Research suggests that during childhood and adolescence, growth hormone (GH) and its physiologic mediator insulin-like growth factor I (IGF-1) promote intestinal Ca absorption in two ways. The first effect is through activation of renal CYP27B1 and the elevation of serum 1,25(OH)_2_ D levels [107]. However, in adult animals, GH treatment increases intestinal Ca absorption without significantly increasing serum 1,25(OH)_2_ D levels [108]. It does so by modulating intestinal VDR levels and increasing cell sensitivity to 1,25(OH)_2_ D [109]. The effect of GH on Ca absorption is likely mediated through IGF-1 but the intestinal actions of IGF-1 may also be independent of vitamin D signaling [49,110].

Dietary Ca requirements increase significantly during the third trimester of pregnancy and during lactation to meet the needs of the fetus and term infant. While pregnancy causes a vitamin D-independent increase in Ca absorption whose mechanism is not clearly understood [111,112,113,114], during late pregnancy serum 1,25(OH)_2_ D levels and intestinal Ca absorption are both elevated [115]. This is because of PTH-independent 1,25(OH)_2_ D production by the placenta [116]. Ca absorption is also regulated during lactation in rodents (but not humans) but this is due to a prolactin-dependent mechanism [117,118]. However, prolactin cooperates with 1,25(OH)_2_D_3_ to regulate intestinal Ca transport and the expression of TRPV6 and calbindin-D_9k_ in rats [119], suggesting prolactin acts together with 1,25(OH)_2_D_3_ to increase active intestinal Ca absorption.

Aging reduces Ca absorption efficiency [120,121,122,123,124,125]. Yet, despite the fact that age-associated Ca malabsorption was discovered 50 ago, we still do not know the molecular mechanism underlying this phenomenon. Lower serum 1,25(OH)_2_D levels in the elderly has been reported in some studies [126,127] but not others [128]. In fact, some research indicates that serum 1,25(OH)_2_ D is higher in older subjects even though fractional Ca absorption is not changed [128,129,130]. Similar age-associated intestinal resistance to 1,25(OH)_2_ D signaling has been formally demonstrated in rats [124] and humans [125]. Some evidence suggests that this phenomenon may be caused by lower intestinal VDR levels [130,131,132] but after adulthood is reached, age-related declines in intestinal VDR content are modest (−20%) [130,131] or non-existent [124]. Consistent with the lack of an impact of age on VDR expression, we recently reported that the open chromatin regions that control the expression of the intestinal VDR gene are not different between 3- and 21-mo-old mice, [104]. Thus, while my research group [53] has shown that a 50% reduction in intestinal VDR level blunts the intestinal response to elevated serum 1,25(OH)_2_D levels, the inconsistency in the reports on the impact of age on intestinal VDR levels suggests that other mechanisms may contribute to age-associated intestinal resistance to vitamin D.

Another aspect of aging that could negatively impact vitamin D metabolism or intestinal regulation of Ca absorption is the decline in sex hormone levels. Consistent with this, estrogen loss severely disrupts Ca metabolism in post-menopausal women, including reducing Ca absorption [133,134]. While estrogen signaling directly regulates intestinal Ca absorption [135,136,137], it also enhances the intestinal responsiveness to 1,25(OH)_2_ D [138]. Some [109,139,140], but not all [141], studies report that low estrogen levels reduce intestinal VDR levels and that this is responsible for intestinal vitamin D-resistance following estrogen loss. In prepubertal boys, testosterone therapy increased intestinal Ca absorption by 61% [142] and this was accompanied higher serum IGF-1 levels that might influence vitamin D metabolism. As men age, both Ca absorption efficiency and serum levels of the sulfated form of the testosterone prohormone DHEA, dehydroepiandrosterone sulphate (DHEAS), fall significantly [143]. However, the change in Ca absorption was independent of changes in serum 1,25(OH)_2_ D, suggesting changes in androgen signaling do not alter vitamin D metabolism. It is not known if testosterone regulates intestinal VDR levels.

## 6. Conclusions

A large amount of data supports the conclusion that adequate vitamin D status and adequate production of the metabolite 1,25(OH)_2_ D are needed to support transcellular, saturable intestinal Ca absorption. 1,25(OH)_2_ D regulates intestinal biology by activating the VDR to stimulate gene expression. This increases the maximum capacity of the Ca transport system by increasing levels of a transporter that mediates saturable, transcellular Ca transport. The saturable, vitamin-D-regulated component of intestinal Ca absorption plays a significant role in maintaining Ca absorption efficiency because in most individuals, serum 1,25(OH)_2_ D levels are elevated by habitually low dietary Ca intake that is common in the general population. The exact mechanism that describes Ca movement through the enterocyte is still in question. The facilitated diffusion model is the best characterized mechanism but there are some inconsistencies in the model that must be resolved through additional research. In contrast, research supports a model for regulated paracellular Ca movement through tight junctions that predominates across the ileum and in the early postnatal period. Overall, the data suggest that vitamin D signaling regulates intestinal Ca absorption by different mechanisms that are segment specific.

## Figures and Tables

**Figure 1 nutrients-14-03351-f001:**
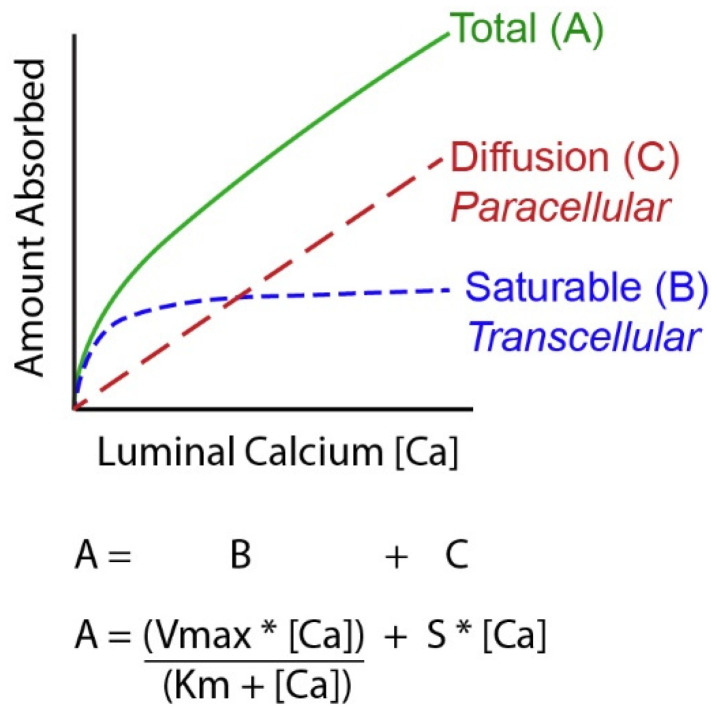
A Mathematical model of intestinal calcium (Ca) absorption. By studying Ca absorption over a range of luminal Ca levels it has been shown that the total amount of Ca absorbed across the intestinal barrier can be described as a curvilinear function. Total transport (A) is the sum of a saturable component (likely transcellular, B) that can be defined by the Michaelis–Menten equation and a diffusional process (C) that is defined by a straight line. [Ca] = luminal Ca concentration; S = the slope of the diffusional component; Vmax = the maximum transport rate seen for the saturable transport component; Km = the luminal concentration of the mineral at ½ the Vmax.

**Figure 2 nutrients-14-03351-f002:**
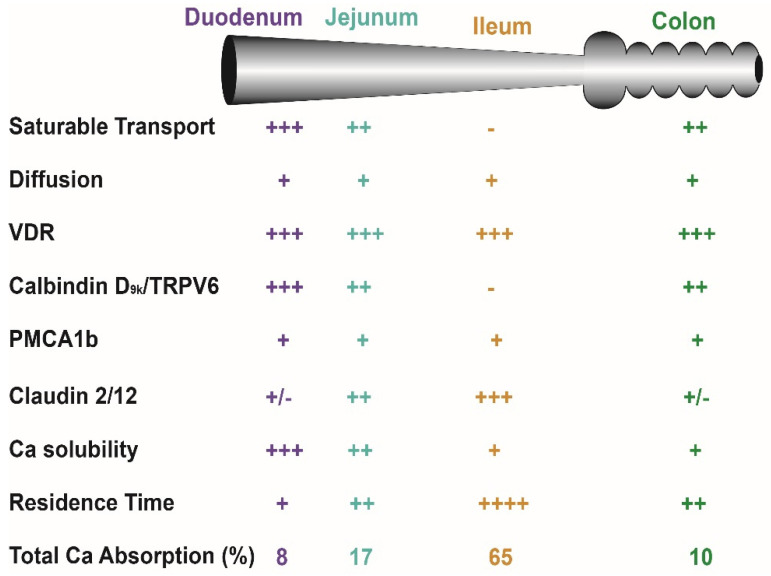
Critical factors that influence intestinal calcium (Ca) absorption. Many factors influence net Ca absorption and distinct Ca absorption mechanisms used in various intestinal segments. See Bronner and Pansu [38] for a discussion of solubility and transit time as factors affecting Ca absorption. The number of “+” signs reflects the magnitude of the parameter across tissues while a “-” sign indicates that the parameter is absent in a segment.

**Figure 3 nutrients-14-03351-f003:**
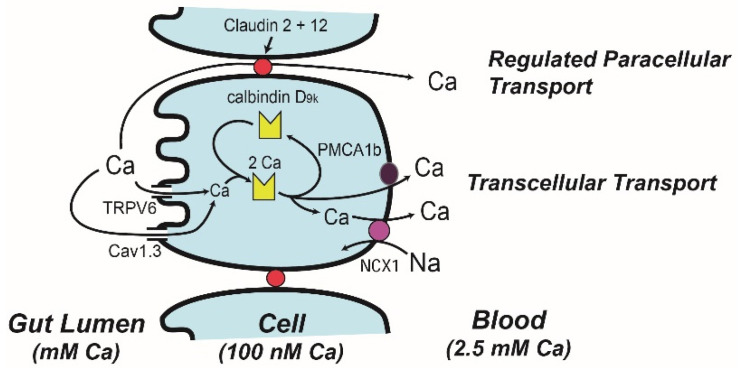
A Model describing intestinal calcium (Ca) absorption. The transcellular absorption pathway is described by the facilitated diffusion model while a regulated paracellular transport mechanisms mediated by claudin 2 and claudin 12 provides selectivity for Ca movement through the tight junction complex. For details of how vitamin D regulates various aspects of these models refer to the text.

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
