# Peer review of "Vitamin D-Mediated Regulation of Intestinal Calcium Absorption"

_nutrients, 2022, doi:10.3390/nu14163351_

Round 1

Reviewer 1 Report

Line 365: This reviewer could not find the explanation for paracellular Ca transport in the early postnatal period in lines 290-350.

Line117-118: The sentence is inappropriate, because "Km" value should be indicated with the unit equivalent to concentration such as mM.

Figure 2 legend: What is Tx?
Line 49: The corresponding "Figure 1" is missing.
Line 58: The "Vmax" needs the explanation described in line 109.
Line 82: Is "This this case" mistyped?
Line 165: "VDR mouse" should be "VDR KO mouse."
Line 261: The "distal intestine" may include the colon, in which claudin-2/12 expression appears to be minimal in Fig.2,

Author Response

  • Line 365: This reviewer could not find the explanation for paracellular Ca transport in the early postnatal period in lines 290-350.

It is not clear what the reviewer is asking. Neither line 365 nor lines 290-350 mention paracellular Ca transport in the early postnatal period.  Is he/she suggesting it should be mentioned?

(2) Line117-118: The sentence is inappropriate, because "Km" value should be indicated with the unit equivalent to concentration such as mM.

I apologize for not fully explaining this point.   The Km is 3.3 mM.  When one “back-calculates” to meal Ca level, this is ~265 mg.  I’ve corrected that in the manuscript.

(3) Figure 2 legend: What is Tx?

Tx has been removed.

(4) Line 49: The corresponding "Figure 1" is missing.

This reference was from an earlier draft and it has been deleted.

(5) Line 58: The "Vmax" needs the explanation described in line 109.

The manuscript already included definitions for Vmax and Km.  “where it increases the Vmax (maximal absorptive capacity) but not Km (the affinity of the process for Ca).”

(6) Line 82: Is "This this case" mistyped?

The repeated word has  been removed.

(7) Line 165: "VDR mouse" should be "VDR KO mouse."

This has been corrected.

(8) Line 261: The "distal intestine" may include the colon, in which claudin-2/12 expression appears to be minimal in Fig.2,

The phrase has been corrected to “distal small intestine”.

Reviewer 2 Report

The review entitled “Vitamin D-Mediated Regulation of Intestinal Calcium Absorption” is very well-written, comprehensive, and interesting to read.

A few suggestions for improvement are:

The text in the paragraph (lines 117-129) may be expanded to include some more explanation to Figure 2, especially with the reference to Calbindin D9k/TRPV6, PMCA1b and Claudin 2/12.

Line180: The role of the Fok I gene should be stated

Typo in line 82

Line 122: add “in” before the word each

References 29 and 37 in lines 125 and 129 respectively can be joined with other references.

Author Response

  • The text in the paragraph (lines 117-129) may be expanded to include some more explanation to Figure 2, especially with the reference to Calbindin D9k/TRPV6, PMCA1b and Claudin 2/12.

Section 4 “Molecular Models of Ca absorption” specifically refers to the proteins in Figure 2.

  • Line180: The role of the Fok I gene should be stated

Fok I is not a gene but a restriction fragment length polymorphism that result upon digestion of DNA with the restriction enzyme Fok I.  This has been changed in the manuscript.

  • Typo in line 82

The duplicated word has been removed.

  • Line 122: add “in” before the word each

This has been corrected.

  • References 29 and 37 in lines 125 and 129 respectively can be joined with other references.

This has been corrected.

Reviewer 3 Report

This is an excellent, well written review.  There has not been such an update in several years.

Fig. 1  Caption  line 65  A word is missing "it has been shown"

Fig. 2  Is very informative and not see in other general reviews.  Do we know that saturable transport machinery is absent in the ileum or just that it has already occurred sooner?

line 250  "has focused on vitamin D-regulated" 

Author Response

Fig. 1  Caption  line 65  A word is missing "it has been shown"

This has been corrected

Fig. 2  Is very informative and not see in other general reviews.  Do we know that saturable transport machinery is absent in the ileum or just that it has already occurred sooner?

Citation 2 is the experimental demonstration that the ileum does not have saturable, vitamin D regulated Ca absorption in rats.

line 250  "has focused on vitamin D-regulated" 

This has been corrected.